# Development of Thrips Repellents, and Their Combined Application with Aggregation Pheromones in a Push–Pull Strategy to Control *Frankliniella occidentalis*

**DOI:** 10.3390/insects16111137

**Published:** 2025-11-07

**Authors:** Xiaowei Li, Yiming Pan, Yunxu Wang, Yaru Wang, Zhijun Zhang, Yaobin Lu

**Affiliations:** 1State Key Laboratory for Quality and Safety of Agro-Products, Institute of Plant Protection and Microbiology, Zhejiang Academy of Agricultural Sciences, Hangzhou 310021, China; lixiaowei1005@163.com (X.L.); pym9248@163.com (Y.P.); m17336369346@163.com (Y.W.); zhijunzhanglw@hotmail.com (Z.Z.); 2Xinjiang Key Laboratory of Agricultural Biosafety, Urumqi 830091, China; 3Xianghu Laboratory, Institute of Bio-Interaction, Hangzhou 311231, China; wangyaru0617@163.com

**Keywords:** western flower thrips, eucalyptol, α-pinene, repellents, push-pull strategy

## Abstract

Insect repellents have been intensively studied due to growing interest in environmentally safe pest control methods. Numerous repellent compounds have been reported to be effective against western flower thrips, *Frankliniella occidentalis*. However, few studies have been conducted to test the repellent activity and control efficacy of these compounds against thrips in the field. In this study, we investigated the repellent effects of two plant volatiles, α-pinene and eucalyptol, on *F. occidentalis* using cage and field experiments. Our results showed that both α-pinene and eucalyptol at high doses significantly reduced the egg-laying of *F. occidentalis* on host plants in cage experiments. In addition, the application of eucalyptol EC and α-pinene EC reduced the field population of *F. occidentalis*, with eucalyptol EC performing better. The combination of aggregation pheromone traps and repellent eucalyptol spray also significantly reduced *F. occidentalis* populations in the field. Our results provided effective eco-friendly control methods for the integrated pest management of *F. occidentalis*.

## 1. Introduction

The western flower thrips, *Frankliniella occidentalis* (Pergande), is a significant worldwide pest of horticultural and agricultural crops [1]. *Frankliniella occidentalis* causes damage through direct feeding and oviposition on leaf tissues, buds, and flowers [2]. With high reproductive capacities and short generation times, this species could reach high infestations, causing significant economic losses. In addition, *F. occidentalis* also causes severe indirect crop damage through the transmission of plant viruses, e.g., tospoviruses [3]. Currently, the application of synthetic insecticides remains the most used control measure for *F. occidentalis* [4,5,6]. However, the intensive use of insecticides has led to the wide development of resistance [7,8,9,10]. Alternative and eco-friendly control methods should be developed within the context of integrated pest management.

Regulating the behavior of pests through plant-released semiochemicals to achieve the purpose of pest control is one important measure in sustainable pest management [11,12,13]. Insect repellents have been intensively studied due to growing interest in environmentally safe pest control methods and the increasing number of insecticide-resistant pest populations [14]. Various studies have reported the behavioral regulatory effects of plant volatiles on thrips pests, but mainly focused on attractants, with limited studies on repellents [15,16,17,18,19]. Very few commercialized thrips repellents are currently available. Several repellent compounds have been reported to be effective against thrips pests [20]. For *F. occidentalis*, a number of compounds have been reported to be repellent based on laboratory olfactometer bioassays, including salicylaldehyde [21], methyl salicylate [22], β-ionone [23], β-ocimene, γ-terpinene, β-phellandrene [24], and linalool [25]. Rosemary (*Rosmarinus officinalis*) is a small shrub of the genus *Rosmarinus* in the Lamiaceae family. Previous studies have shown that rosemary is repellent against various pests such as whiteflies [26], aphids [27,28,29], tea geometrid [30], and leafhoppers [31]. Our previous study found that rosemary has repellent effects on three thrips species, including *F. occidentalis* (Pergande), *Frankliniella intonsa* (Trybom), and *Thrips palmi* Karny [32]. A total of 47 compounds were identified from the volatiles of rosemary, with α-pinene and eucalyptol being the most abundant compounds [32]. The two compounds, α-pinene and eucalyptol, showed significant repellent activities for two *Frankliniella* thrips species in a Y-tube olfactometer [32]. However, it remains unclear whether these two compounds could repel *F. occidentalis* when sprayed on plants, and whether they can be further developed into thrips repellents for *F. occidentalis* control in the field.

The push–pull strategy is a useful tool for integrated pest management and has been widely used in the management of many pests [33,34]. Attractant–repellent combinations have been reported as an effective push–pull strategy for thrips control [35,36]. In this strategy, repellent compounds were used as a “push” component to repel thrips from crops or prevent damage on crops, while attractants were used as a “pull” component to mass-trap thrips. Attractants, including pheromones and plant volatiles, have been intensively reported for *F. occidentalis* [20]. Aggregation pheromone compounds of *F. occidentalis*, neryl (*S*)-2-methylbutanoate and (*R*)-lavandulyl acetate, which were produced by adult male thrips and attract both females and males, have been identified and widely used for *F. occidentalis* monitoring and mass-trapping [37,38,39,40]. Exploring the potential combined use of aggregation pheromone attractants and plant repellents could provide a practical control strategy for this pest.

In this study, in order to develop repellents and an attractant–repellent push–pull strategy for *F. occidentalis*, we first investigated the effects of α-pinene and eucalyptol at different doses on their oviposition on pepper plants in cage experiments. Then, we further evaluated the control efficiency of repellents on *F. occidentalis* in pepper fields. Finally, the control efficiency of aggregation pheromone attractants–plant volatile repellents combinations was investigated.

## 2. Materials and Methods

### 2.1. Test Insects and Plants

An *F. occidentalis* colony was collected on *Cucumis melo* plants in a greenhouse in Beijing, China. The collected thrips colony was maintained on *Phaseolus vulgaris* L. bean pods in glass jars in climate rooms (25 ± 1 °C, 65% ± 5% RH, 16L:8D photoperiod) for more than 10 years. Female adults (1 to 5 days old) from the colony were used for cage egg-laying experiments.

Seeds of pepper (Hangjiao No. 1, Hangzhou Sanjiang Seeds Co., Ltd., Hangzhou, China) were sown in coconut coir for germination. After reaching the 2~3 leaf stage, pepper seedlings were transplanted into plastic pots (7 cm diameter and 9 cm height, Hangzhou Anhua Plastic Industry Co., Ltd., Hangzhou, China) and placed in insect-free cages in a greenhouse (27 ± 1 °C, 70% ± 5% RH, 16L:8D photoperiod). Plants at the 7–10 leaf stage were used for cage egg-laying experiments.

### 2.2. Chemicals

The repellent compounds, eucalyptol (purity ≥ 98%) and α-pinene (purity ≥ 98%), were purchased from Aladdin Biochemical Technology Co., Ltd. (Shanghai, China). The 20% eucalyptol and α-pinene EC formulations were prepared as follows: 20% eucalyptol or α-pinene + 10% Tween 80 + 70% ethanol by volume.

The two *F. occidentalis* aggregation pheromone compounds, neryl (*S*)-2-methylbutanoate (at least 98% purity, 99% enantiomeric excess) and (*R*)-lavandulyl acetate (at least 98% purity, 98% enantiomeric excess), were synthesized at China Agricultural University according to the method described by Hamilton et al. (2005) [37]. The pheromone lures for field attraction were prepared as follows: 1250 ng (*R*)-lavandulyl acetate and 10,000 ng neryl (*S*)-2-methylbutanoate were blended in hexane, and loaded in rubber septa (1 cm diameter × 1.5 cm long, pre-cleaned with hexane) [41].

### 2.3. Oviposition Selection in Cage Experiments

The experiments were conducted in 300-mesh insect cages (50 cm × 50 cm × 50 cm, Ningbo Jiangnan Instrument Factory, Ningbo, China). Since 1 μL was the most repellent dose for eucalyptol and α-pinene against *F. occidentalis* in the Y-tube olfactometer [32], the doses for cage experiments started from 1 μL. To achieve the effective component concentrations of 1, 10, 100, and 200 μL/mL, respectively, 20% eucalyptol EC and 20% α-pinene EC were diluted with water. Each treated pepper plant was sprayed with 1 mL of repellent solution at different concentrations using 2 mL spray bottles. The control plant was sprayed with 1 mL of water. The control and treated plants were placed in opposite corners of the cage. Since the age of thrips affect their reproductive performance, female adult thrips with similar ages (3–5 days old) were selected from the colony to ensure similar reproductive performance of thrips for all the treatments. Sixty female adult thrips were released into the middle of the cage and allowed to spread freely. To avoid the effect of environmental factors on the reproductive performance of thrips, the cages were placed in climate rooms at the same conditions as the thrips colony (25 ± 1 °C, 65% ± 5% RH, 16L:8D photoperiod). After 48 h, the number of eggs laid on the host plants in each treatment was recorded under a transmitted light microscope. The experiment was repeated 6 to 9 times for each compound at each concentration. The percentage of eggs on control or treated plants was calculated by dividing the total number of eggs laid on control or treated plants by the total number of eggs laid on both the control and treated plants across replicates for each dose treatment.

### 2.4. Control Efficacy in Field Trials

Field trials were carried out in a pepper greenhouse (45 m × 8 m). The pepper plants were grown in the ground in an unheated high tunnel. The pepper plants were at the early flowering stage and infested with a low level of *F. occidentalis*. The experimental area in the greenhouse was divided into 9 plots. Plots (1 m × 10 m) were 1 m apart from each other. Three treatments (control, eucalyptol treatment, and α-pinene treatment) were randomly arranged, with three plots per treatment. For the eucalyptol and α-pinene treatments, 1 mL of 200 μL/mL eucalyptol EC, and 1 mL of 100 μL/mL α-pinene EC were sprayed on each plant, respectively. For the control treatment, 1 mL of water was sprayed on each pepper plant. The number of *F. occidentalis* adults on pepper flowers was recorded by visual inspection before application and on the 1st, 2nd, and 3rd day after application, respectively. In each plot, 6~10 plants were randomly selected, and the number of thrips on all the flowers was recorded.

The control efficacy was calculated for eucalyptol treatment and α-pinene treatment using the following equations [42]:Population reduction rate (%) = (number of thrips per flower before treatment − number of thrips per flower after treatment)/number of thrips per flower before treatment × 100%.Control efficacy (%) = (population reduction rate in repellent application treatment − population reduction rate in control treatment)/(1 − population reduction rate in control treatment) × 100%

### 2.5. Push–Pull Strategy with Thrips Repellents and Aggregation Pheromone Lures

The control efficacy of a push–pull strategy (the combined uses of thrips repellents and aggregation pheromone lures) was investigated in different pepper greenhouses. In the push–pull strategy greenhouse, the thrips repellent with high control efficacy, based on the results from 2.4, was sprayed as described above. Meanwhile, forty aggregation pheromone lures were stuck in the middle of rectangular blue sticky traps (20 cm × 25 cm) and hung evenly in the greenhouse, with the base at about 10 cm above crop height. In the repellent-only greenhouse, thrips repellent was sprayed without the application of aggregation pheromone lures. In the control greenhouse, no control measures were applied. The number of *F. occidentalis* adults on pepper flowers was investigated using the five-point sampling method before treatment and on the 1st, 2nd, and 3rd days after application, respectively. At each point, 10 plants were randomly selected, and the number of *F. occidentalis* adults on all the flowers was recorded.

### 2.6. Data Analysis

The data were analyzed using the SPSS software Version 16.0 (SPSS Inc., Chicago, IL, USA). The data from the cage experiments were analyzed using the chi-square test, assuming a ratio of 1:1 between the two treatments. Control efficacies between the treatments in the field were compared using repeated-measure ANOVAs. Pairwise comparisons were performed with the LSD test (*p* < 0.05).

## 3. Results

### 3.1. The Effects of Repellent Sprays on F. occidentalis Oviposition

Spraying eucalyptol significantly influenced the oviposition behavior of *F. occidentalis* at different doses (Figure 1). Compared to the control pepper plants, no significant differences were found in the percentage of eggs laid on pepper plants sprayed with eucalyptol at doses of 1 μL, 10 μL, or 100 μL (1 μL: *χ*^2^ = 2.560, *df* = 1, *p* = 0.110; 10 μL: *χ*^2^ = 1.440, *df* = 1, *p* = 0.230; 100 μL: *χ*^2^ = 3.240, *df* = 1, *p* = 0.072). However, *F. occidentalis* laid significantly fewer eggs on pepper plants sprayed with 200 μL eucalyptol (*χ*^2^ = 73.960, *df* = 1, *p* < 0.0001), with only 6.57% of eggs laid on plants treated with 200 μL eucalyptol compared to the control plants.

Spraying α-pinene also significantly influenced the oviposition behavior of *F. occidentalis* at different doses (Figure 2). Compared to the control pepper plants, no significant differences were found in the percentage of eggs laid on pepper plants sprayed with 1 μL or 10 μL α-pinene (1 μL: *χ*^2^ = 0.360, *df* = 1, *p* = 0.549;10 μL: *χ*^2^ = 1.960, *df* = 1, *p* = 0.162). However, *F. occidentalis* laid significantly fewer eggs on pepper plants sprayed with 100 μL α-pinene (*χ*^2^ = 40.960, *df* = 1, *p* < 0.0001), with only 17.90% of eggs laid on plants treated with 100 μL α-pinene compared to the control plants.

### 3.2. Control Efficacy of Repellent Sprays in Field Trials

Field results showed that the application of eucalyptol and α-pinene could significantly reduce the population of *F. occidentalis* in the field (Figure 3). Although no significant difference was found between the control efficacy of eucalyptol and α-pinene (*F*_1,4_ = 4.435, *p* = 0.103), the control efficacy of eucalyptol was relatively higher, with an average control efficacy of 70.90% (range from 52.86%~94.18%), 80.96% (range of 70.55%~91.60%), and 67.59% (range of 61.23~84.09%) on day 1, day 2, and day 3, respectively. The control efficacy of α-pinene was 66.66% (range of 56.32%~76.30%), 50.54% (range of 31.38%~80.90%), and 49.71% (range of 33.95%~69.10%) on day 1, day 2, and day 3, respectively. Consequently, eucalyptol performed better as a thrips repellent, with the highest control efficacy of 80.96%.

### 3.3. Control Efficacy of Push–Pull Strategy with Thrips Repellents and Aggregation Pheromone Lures

Field results showed that, compared to the control greenhouse, both the push–pull strategy and repellent application inhibited the population increase in thrips (Table 1). In the push–pull strategy, the average control efficacy was 81.95% (range of 58.65%~97.40%), 71.08% (range of 45.22%~99.00%), and 79.90% (range of 45.72%~92.65%) on day 1, day 2, and day 3, respectively (Table 1). In the repellent application greenhouse, the average control efficacy was 77.70% (range of 55.41%~94.52%), 75.07% (range of 66.90%~89.56%), and 70.30% (range of 60.99%~87.26%) on day 1, day 2, and day 3, respectively (Table 1). The control efficacies in the push–pull strategy greenhouse on day 1 and day 3 were higher than those in the repellent application greenhouse, although no significant difference was found between the two treatments based on repeated-measure ANOVAs (*F* = 0.070, *p* = 0.798).

## 4. Discussion

Our previous study reported that eight of the ten most abundant volatile compounds in rosemary leaves, α-pinene, eucalyptol, camphene, borneol, β-pinene, D(+)-camphor, β-myrcene, and bornyl acetate, showed significant repellent effects on *F. occidentalis* [32]. The two most abundant compounds from rosemary volatiles, α-pinene and eucalyptol, also showed strong repellent activity against other thrips species [32]. However, few studies have been conducted to test the repellent activity and control efficacy of repellents on thrips in the field. In this study, we investigated the repellent effects of α-pinene and eucalyptol on *F. occidentalis* using cage and field experiments. Our results indicated that both α-pinene and eucalyptol at high doses (100 μL of α-pinene and 200 of μL eucalyptol, respectively) significantly reduced the percentage of eggs laid by *F. occidentalis* on host plants in cage experiments. The repellent doses in cage oviposition experiments were much higher than those in a Y-tube olfactometer, which ranged from 0.01 to 1 μL for α-pinene and 0.1 to 1 μL for eucalyptol [32]. This discrepancy might be due to the larger space for volatile diffusion and the more complicated background odors in cages compared to the Y-tube olfactometer. Nevertheless, both cage experiments and Y-tube olfactometer demonstrated that α-pinene repelled *F. occidentalis* at lower doses compared to eucalyptol. Conversely, the application of eucalyptol EC and α-pinene EC could reduce the field population of *F. occidentalis*, with eucalyptol EC performing better. The differing results of the two compounds in the field may be due to the distinct responses of the two compounds to background odors (additional environmental volatile cues). It has been reported that background odors from crop plants influence the response of insects to volatile cues in their habitat [43,44], affecting the efficacy of these compounds in field applications. Moreover, the complex biotic and abiotic factors present in field conditions also impact the results. Further research is needed to determine how different factors influence the repellent activities of these two compounds.

The biological activities of α-pinene against insects have been widely reported [45], including sanitary insects [46,47], stored product insects [48], and forest insects [49]. However, there have been only a few studies on its effects on agricultural insects. The biological activities of α-pinene against insects included repellent, feeding-deterrent, oviposition-deterrent, and insecticidal activities [50]. The repellent activity of α-pinene against *F. occidentalis* was demonstrated in our previous study [32], while the results from cage experiments in this study showed that it also had oviposition-deterrent activities. The oviposition-deterrent activity reported here is consistent with the results for the oriental fruit fly, *Bactrocera dorsalis* [51]. Eucalyptol has been reported to have strong repellent activity against insect species, including whiteflies [52], aphids [27], and tomato leafminers [53]. The repellent activities of eucalyptol against thrips, including *F. occidentalis,* were reported in our previous study [32]. Similarly, Tsuro et al. (2025) [54] demonstrated that transgenic torenia (*Torenia fournieri* Lind.) plants emitting eucalyptol exhibited significantly enhanced repellency against *F. occidentalis* compared to non-transgenic plants. The effect of eucalyptol on oviposition and field insect population has also been reported for the cabbage root fly, *Delia radicum*, where field application of eucalyptol reduced oviposition on target crops by 45% and reduced the final infestation by *D. radicum* [55], which was consistent with the present study. In addition to repellent effects, some semiochemicals also possess deterrent activities that inhibit feeding or oviposition. Results from both previous [32] and current studies confirm the repellent and oviposition-deterrent effects of α-pinene and eucalyptol. Further research should evaluate whether these two compounds also have feeding-deterrent effects on *F. occidentalis* to better understand their behavioral influence on thrips. Although we demonstrated that field applications of α-pinene and eucalyptol could reduce the population of *F. occidentalis* on target crops, the efficacy was only assessed over 3 days. Future studies are necessary to determine whether the application of repellents could continuously suppress the *F. occidentalis* populations in the field.

The push–pull strategy using thrips attractants and repellents has been reported in thrips management. For instance, a push–pull strategy using the alarm pheromone as the “push” component and the aggregation pheromone as the “pull” component significantly reduced the *F. occidentalis* population on hot pepper plants in the greenhouse [36]. In another study, the combination of the thrips attractant ethyl iso-nicotinate and the repellent *Origanum majorana* essential oil significantly reduced the number of onion thrips, *Thrips tabaci*, on white sticky-plate traps [35]. In our study, we found that the combination of aggregation pheromone traps and repellent eucalyptol spray significantly reduced *F. occidentalis* populations in the field. These studies suggested that the attractant–repellent combination is an effective push–pull strategy for thrips control, and could be an important component in integrated pest management for thrips pests.

In conclusion, this study demonstrates that eucalyptol and α-pinene significantly reduced the percentage of eggs laid by *F. occidentalis* on plants in cage experiments. The spraying of the α-pinene and eucalyptol EC greatly reduced the population of *F. occidentalis* in the field, with eucalyptol EC performing better. A push–pull strategy combining aggregation pheromone traps and repellent eucalyptol spray was an effective strategy to suppress the *F. occidentalis* field population. The repellents and push–pull strategy developed in this study provide effective eco-friendly control methods for the integrated pest management of *F. occidentalis*.

## Figures and Tables

**Figure 1 insects-16-01137-f001:**
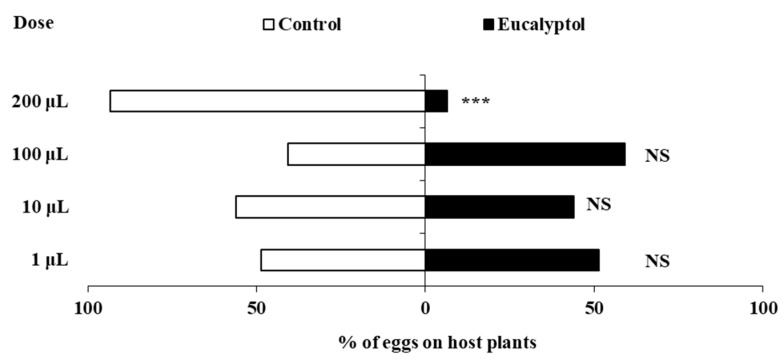
The effects of eucalyptol spray on the oviposition of *F. occidentalis* females. Asterisks indicate significant differences within a choice test (*** *p* < 0.001) and NS indicates no significant difference.

**Figure 2 insects-16-01137-f002:**
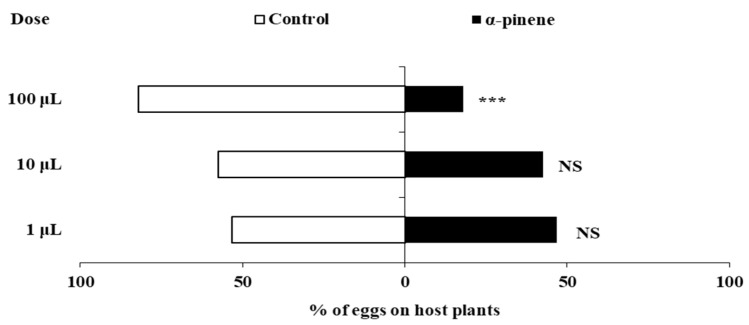
The effects of α-pinene spray on the oviposition of *F. occidentalis* females. Asterisks indicate significant differences within a choice test (*** *p* < 0.001) and NS indicates no significant difference.

**Figure 3 insects-16-01137-f003:**
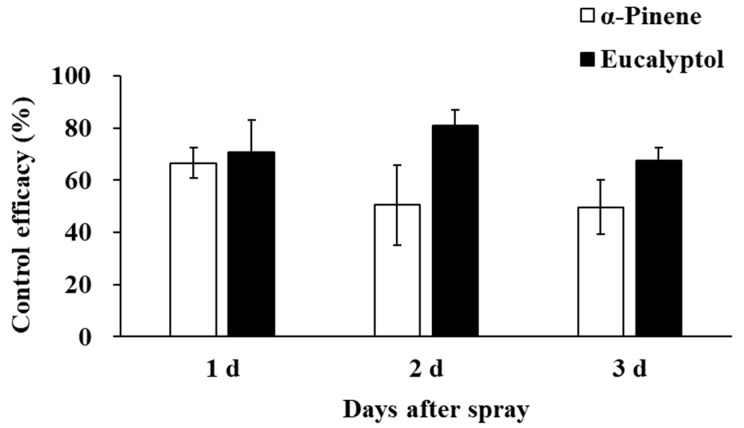
The control efficacy of eucalyptol and α-pinene on *F. occidentalis* populations in the field.

**Table 1 insects-16-01137-t001:** Field control efficacy of push–pull strategy on *F. occidentalis* in the field.

Greenhouses	Initial Number per Flower	Number of Thrips per Flower			Control Efficacy (%)		
		Day 1	Day 2	Day 3	Day 1	Day 2	Day 3
Push–pull strategy	0.33 ± 0.11	0.28 ± 0.11	0.35 ± 0.09	0.39 ± 0.10	81.95 ± 7.06	71.08 ± 10.41	79.90 ± 8.52
Thrips repellent only	0.71 ± 0.20	1.00 ± 0.76	1.29 ± 0.74	1.72 ± 0.54	77.70 ± 6.71	75.07 ± 4.64	70.30 ± 4.62
Control	0.23 ± 0.07	1.33 ± 0.20	1.82 ± 0.64	1.34 ± 0.20	-	-	-

## Data Availability

The data presented in this study are fully contained within the figures of the article. Further inquiries can be directed to the corresponding author.

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
