# Peer review of "Development of Thrips Repellents, and Their Combined Application with Aggregation Pheromones in a Push–Pull Strategy to Control *Frankliniella occidentalis"

_insects, 2025, doi:10.3390/insects16111137_

Round 1

Reviewer 1 Report

Comments and Suggestions for Authors

The present study illustrated the repellent effects of plant compounds on thirps performance, however, there were some important issues need to be addressed.  
1. The authors using the limited parameter, e. g. oviposition behavior, to illustrate the repellent effects, it is not sufficient to point that. Personally suggestion: adding the feeding preferrance tests, or plant damage sacle descriptation, etc.
2. The discussion section lacked the discussion in depth. The first para could be placed into the introdction, the reasons why spraying the 100ul Eucalyptol EC induced the thirps reproduce more eggs than 1 and 10ul?  etc.

Specific issues:
1. Not mention:How to guarantee biotic and abiotic stress won't exhibited the effects on the performance of test insects, that direct collect from greenhouse.
2. How to avoid the negative effects of host transferance on been pods raising thirps, when they infested on the pepper?
3. The principals of the concentration selection of chemicals treatment need to be illustrated.
4. Chi test didn't point out in data analysis section. Because the 3.1 section using the Chi test.
5. Why the precent of eggs laying in controls are exhibited the huge differentiation?
6. You should point out how to caluate the % of eggs laid on host plants.
7. you should mark the letters of differences between different treatments.

Minor issues: the latin names of insects or plants should be italized.
L239-240 Uncomplete sentence; Plant expressing eucalyptol?

Reviewer 2 Report

Comments and Suggestions for Authors

The manuscript was about the “Development of thrips repellents, and their combined application with aggregation pheromones in a push-pull strategy to control Frankliniella occidentalis”. The authors investigated the repellent effects of two plant volatiles, α -pinene and eucalyptol, using cage and field experiments. Results showed that both α -pinene and eucalyptol at high doses significantly reduced the egg laying of the thrips on host plants in cage experiments and the application of eucalyptol EC and α -pinene EC reduced the field populations. It contains useful information for thrips IPM, however, it needs major revisions before acceptance. There are several issues with grammar. The methods need to be more elaborate, specifically, with addition of details like pot sizes, sources of materials etc.  The biggest issue is that F. occidentalis lays eggs primarily in leaf tissues. The methods state eggs were counted on leaf surfaces. This needs an explanation as I question the validity of the experiment as explained. Several suggested changes are found as comments within the attached pdf file.

Round 2

Reviewer 1 Report

Comments and Suggestions for Authors

Although the authors had been resolved most of issues in the revision, except comments 3 and 4, personal feels that the data get from the direct collected field population of thrips is not sufficient for concluding the conclusion. Because this is an Insect journal, and the main topic of this manuscript is to illustrate the control strategy of thrips, it is better to using the experimental raising colony instead of directly adopting the field colonies.  Therefore, the manuscript is still need to carefully treat or point out the following points for the next revision. I strongly suggested the authors to provide more information about why directly using the field colony to conduct the experiment, or it is hard to raise the field thrip populations a couple of generation prior to infestation experiment.  

Reviewer 2 Report

Comments and Suggestions for Authors

I thank the authors for their revisions that clarified the manuscript. There are still a few minor edits that need to be made (grammar and other things). Other than those, it looks acceptable to publish after they are addressed. See pdf file for comments.
